# A French Preoperative Cholesteatoma Management: Current Preoperative Consultation and Tendencies

**DOI:** 10.3390/jcm13185651

**Published:** 2024-09-23

**Authors:** Benjamin Reliquet, Mireille Folia, Paul Elhomsy, Serge Aho-Ludwig, Caroline Guigou

**Affiliations:** 1Department of Otolaryngology-Head and Neck Surgery, Dijon University Hospital, 21000 Dijon, France; benjamin.reliquet@chu-dijon.fr (B.R.); mireille.folia@chu-dijon.fr (M.F.); 2Anesthesiology and Critical Care Department, Dijon University Hospital, 21000 Dijon, France; paul.elhomsy@chu-dijon.fr; 3Department of Epidemiology and Hospital Hygiene, Dijon University Hospital, 21000 Dijon, France; ludwig.aho@chu-dijon.fr; 4ICMUB Laboratory, UMR CNRS 6302, University of Burgundy, 21000 Dijon, France

**Keywords:** otology, cholesteatoma management, current clinical practices, surgical risks

## Abstract

**Objectives**: This study aimed to characterize the information delivery during preoperative consultations for cholesteatoma removal surgery in 2024. The secondary objective was to identify any factors influencing the information delivered. **Methods**: This study was a practice survey which included 33 closed-ended questions and 1 open-ended question. Seven questions concerned the participants’ characteristics and 2 questions concerned the physiopathology of cholesteatoma. Nine questions focused on surgical information, six questions focused on the procedure modalities and ten questions focused on the risks of complications from the intervention. **Results**: Eighty-two surgeons answered the survey. In 75% of the cases, an information form written by a professional society was provided. The risk of recurrence or residual post-operative cholesteatoma was systematically stated in 78% of cases (n = 64), while the risk of aesthetic sequelae was only stated in 1% (n = 1). Participants working in a university hospital were more likely to inform patients about the risks of vertigo (*p* = 0.04), aesthetic risks (*p* = 0.04), poor functional outcomes (*p* = 0.04), surgical revision (*p* = 0.05) and the risk of peripheral facial paralysis (*p* = 0.05). Surgeons who mainly practiced otology were more likely to inform patients about the risks of recurrence and/or residual cholesteatoma (*p* = 0.02) and taste disturbances (*p* = 0.02). **Conclusions**: Cholesteatoma surgery was well explained to patients during the preoperative consultation, mostly with written support, even if the information given was not the same for all complication risks. It could be useful to create an information form dedicated to cholesteatoma surgery to improve comprehensive information and maintain a trustworthy relationship with patients.

## 1. Introduction

Chronic otitis media (COM) is a common condition, affecting 1 to 46% of the population depending on the country [1,2]. COM encompasses various pathophysiological entities, the most severe and advanced form being cholesteatoma. Cholesteatoma accounts for approximately 5% of chronic otitis media cases, affecting around 5 million patients [3]. The standard treatment for cholesteatoma is surgery, although the approach and methods of removal vary from one surgical team to another [4,5,6,7,8].

Cholesteatoma removal surgery is a delicate procedure with several risks, including hearing deterioration (16% risk for initial surgery with an intact stapes, and an increase to 61% in cases of recurrence involving the stapes), vestibular damage (7% risk of labyrinthine fistula), facial paralysis (2% risk in the immediate postoperative period) and the risk of cholesteatoma recurrence or residual disease in the medium to long term (approximately 5%, depending on the series) [9,10,11,12,13]. Considering the various surgical complications and the nature of the pathology, it is essential to provide the patient with the most objective and comprehensive information during the preoperative consultation. Furthermore, tailoring the discussion to each patient’s level of understanding and specific condition is essential for effective communication.

To improve patient information, the French Society of Oto-Rhino-Laryngology and Head and Neck Surgery (SFORL) has published over thirty informational forms for patients that surgeons can provide alongside consultations. These forms offer a written, consensual and formalized support detailing the procedure, postoperative modalities and specific risks of each surgery. This form is not in itself compulsory. There is no legal provision for it, nor any obligation to provide it. This information form is part of a “good practice” recommended to satisfy the duty to inform set out in the French Public Health Code (Art. L1111-2 and R4127-35) and Code 35 of medical ethics. This notion of the duty to inform differs from one specialty to another, depending on the current state of the art, and simply enables judges to verify, in the event of a liability claim, whether the patient has given informed consent or whether the doctor has committed a fault (in the civil sense) by failing to inform the patient sufficiently. This information form does not replace the information given orally during the consultation but provides the patient with a written record, enabling him or her to reflect after the consultation. These documents are freely available at https://campusorl.fr/public/interventions-chirurgicales-en-orl/ (accessed on 20 September 2024). A description of cholesteatoma surgery and its risks is included in the informational form for chronic otitis media in both adults and children.

In fact, the practice of otology is not exempt from the increasing legal actions in our profession [14,15], as patients continually seek answers regarding both the presumed diagnosis and the proposed treatments. Depending on the country, 3 to 22% of litigation after otologic surgery is due to a lack of preoperative information [16,17].

To our knowledge, no studies have investigated how surgeons deliver information to patients during preoperative consultations for cholesteatoma removal surgery. We aimed to investigate the information provided during preoperative consultations for cholesteatoma removal surgery in France, as there was no standardized information form. To address this, we developed a survey to evaluate the information provided to the patient in the preoperative consultation, ensuring they can give informed consent.

The main objective of this study aimed to describe the French professional clinical practices regarding the information delivery during preoperative consultations for cholesteatoma removal surgery in 2024 through data collected by a nationally distributed survey. The secondary objective was to identify any factors that could influence the information given to patients.

## 2. Materials and Methods

This study was a practice survey. It was performed in accordance with the principles of good clinical practice, and the need for Ethics Committee approval was waived. ENT were informed and consented to their participation in this study. After the examination of this study, it was determined that this trial is outside Jardé’s law field. All respondents to the survey were included. There were no exclusion criteria.

An anonymous electronic survey was created and distributed using the GoogleForms platform (Google, Mountain View, California, USA) between October 2023 and March 2024 according to CHERRIES recommendations [18]. It was sent to the professional email addresses of otology professors from French university hospitals to increase local dissemination. The survey was published in an SFORL newsletter accessible to all its members and on an internet platform for ENT (Ear, Nose and Throat) specialists in February 2024.

The survey consisted of 34 questions (33 closed-ended multiple-choice questions and one open-ended question). 

-Seven questions concerned the participants’ anonymized personal information (gender, age, number of surgeries performed per year, years of practice, geographic area, mode of practice, place of practice).-Two questions detailed the initial consultation and the discussion with the patient about the cholesteatoma pathology:How do you provide the information on the physiopathology of cholesteatoma? There are four possible answers (orally, with preoperative imaging, with a diagram, and no). Do you explain the risks of expectative management? There are five possible responses: yes, with the risk of superinfections; yes, with the risk of hearing damage; yes, with the risk of local aggression; only if the patient asks me; no. 

-Nine questions concerned the surgical information (including the open-ended question):Do you ask the patient what he or she expects from the surgery? There are two possible responses: yes or not.How do you provide preoperative information forms? There are four possible responses: yes, an information form authored by a professional society; yes, an information form locally drafted; yes, and the form is countersigned in the patient’s medical file; no.The open-ended question: “Why do you not provide an information form about the surgery?”, which was asked only if participants answered negatively to the question about giving an information form to the patient during the preoperative consultation.Then, a list of six questions were asked about the surgery: Do you show the size of the retro auricular scar? Do you mention a cartilage graft harvested locally? Do you highlight the type of prosthesis used for reconstruction? Do you explain the possibility of mastoid drilling? Do you emphasize facial nerve monitoring? Do you not detail, explaining that the procedure will be adapted based on the course of the surgery?For each of the questions, the possible choices were never, rarely, often and systematically.

-Six questions were about the procedure modalities:Do you state the duration of hospitalization? Do you state the duration of work leave? Do you explain that the functional result is not immediate? Do you detail post-operative care (packing, drops, etc.)? Do you outline the necessary follow-up appointments? Do you discuss the control imaging?Four responses were possible: never, if the patient asks, often and systematically.

-Ten questions focused on the risks of adverse outcomes related to the procedure:Do you explain the risks of facial paralysis, poor functional outcome, new intervention, vertigo, tinnitus, aesthetic sequelae, recurrence or residual, taste disturbances, meningitis and deep abscess, delayed healing and dehiscence? Five responses were possible: never, rarely, only if the situation is appropriate, often and systematically.

Answering any question was not mandatory. All questions were multiple-choice, except for the open-ended question. It took approximately 3 min to complete all the questions.

### Statistical Analysis

Quantitative variables were presented as the median and interquartile range (IQR), while categorical variables were expressed as numbers and percentages. Given the possibility of selecting multiple options for some questions, some percentages exceeded 100%. The tables indicated which questions allowed for multiple responses. Since none of the responses was mandatory, some questions were not answered, resulting in percentages that could be less than 100%. These percentages were calculated based on the number of participants rather than the total number of responses per question.

Statistical analysis was performed using Jamovi software (Version 2.3.28.0, Sydney, Australia). Factors influencing the decision of explaining the risks of complications from the cholesteatoma intervention and the methods of information delivery were investigated. The explanatory variables used were (1) geographical region of practice, (2) age, (3) gender, (4) professional structure, (5) type of professional activity and (6) duration of professional practice.

The chi-squared test or Fisher’s exact test was used to compare categorical variables. Covariates with a *p*-value less than 0.2 in the univariate analysis were considered in the multivariate model [19]. To avoid overfitting, a multivariate analysis was performed with a limited number of predictors [20]. A “*p*” value less than 0.05 was considered statistically significant.

## 3. Results

### 3.1. Population

Eighty-two surgeons have responded to the survey: 35 females and 47 males (Table 1). Forty-five percent (n = 37) of the participants were under 40 years old (Table 1). ENT specialists working in university hospitals are not necessarily those who perform the most cholesteatoma removal surgeries (*p* = 0.1, chi-squared test).

The survey was completed in all regions of France except for Pays de la Loire and Corsica (Figure 1). The most represented region was Île-de-France, with 29% of participants. Six respondents did not specify their geographical location. This may be associated with international respondents who are members of SFORL.

Sixty-two percent of participants worked in a university hospital. All participants had an otology practice, and 52% had an almost exclusive otology practice. Despite this result, only 24% of participants performed more than 40 otologic surgeries per year.

### 3.2. Information on the Cholesteatoma Pathology and the Necessity of Surgical Intervention

During the preoperative consultation for cholesteatoma surgery, the information on the physiopathology of cholesteatoma was provided orally in 77% of cases, with preoperative imaging in 55% of cases and a diagram in 53% of cases, and was not provided in 4% of cases.

The risks of conservative management (not undergoing surgical intervention) were not mentioned in 3% of cases. When discussing the risks of not intervening, those mentioned were superinfections in 77% of cases, hypoacusis in 82% of cases and local aggressiveness in 82% of cases.

### 3.3. Information on the Surgical Procedure (Table 2)

Preoperative information forms were provided in 83% of cases. The information form authored by a professional society was provided in 75% of cases. In 8% of cases, it was locally drafted, and in 18% of cases, it was countersigned in the patient’s medical file.

When no information form was provided (13% of cases; 11 participants; answers of the open-ended question), the reasons were:-The information form was provided by non-medical staff in one case.-The feeling that the form was unnecessary or useless in seven cases.-The inadequacy of the form with the practice’s specificity in one case.-Two participants did not specify their reason.

Sixty-four percent of the participants did not find out what patients expected from cholesteatoma removal surgery.

The details of the surgical procedure were most often explained “systematically” with responses to questions in 43% to 79% of cases. Only 1% of participants never detailed the surgical procedure and explained that it would adapt according to the surgery procedure (Table 2).
jcm-13-05651-t002_Table 2Table 2Surgical Procedure Information. For each question, multiple choices were possible. Percentage calculated based on the number of participants. Percentages > 50% are in bold.QuestionsNeverN (%)RarelyN (%)OftenN (%)SystematicallyN (%)Shows the size of the retro auricular scar 5 (6%)20 (24%)25 (30%)35 (43%)Mentions a cartilage graft harvested locally1 (1%)0 (0%)19 (23%)**65 (79%)**Highlights the type of prosthesis used for reconstruction3 (4%)23 (28%)25 (30%)35 (43%)Explains the possibility of mastoid drilling12 (15%)20 (24%) 16 (20%)37 (45%)Emphasizes facial nerve monitoring9 (11%) 10 (12%)22 (27%) **45 (55%)**Does not detail, explaining that the procedure will be adapted based on the course of the surgery**46 (56%)**22 (27%)3 (4%) 1 (1%)

### 3.4. Information on the Procedure Modalities (Table 3)

Peri- and postoperative modalities were rarely not stated, with 63% to 88% of “systematically” responses to the different questions asked in the survey.

**Table 3 jcm-13-05651-t003:** Surgical Modalities Information. For each question, multiple choices were possible. Percentage calculated based on the number of participants. Percentages > 50% are in bold.

Questions	Never N (%)	If the Patient Asks N (%)	Often N (%)	Systematically N (%)
I state the duration of hospitalization	1 (1%)	2 (2%)	10 (12%)	**72 (88%)**
I state the duration of work leave	3 (4%)	6 (7%)	20 (24%)	**55 (67%)**
I explain that the functional result is not immediate	0 (0%)	4 (5%)	25 (30%)	**55 (67%)**
I detail post-operative care (packing, drops, etc.)	0 (0%)	2 (2%)	16 (20%)	**66 (80%)**
I outline the necessary follow-up appointments	0 (0%)	5 (6%)	22 (27%)	**58 (71%)**
I discuss the control imaging	1 (1%)	12 (15%)	20 (24%)	**52 (63%)**

### 3.5. Information on the Risks of Adverse Outcomes Related to the Procedure (Table 4)

Some risks of complications from the operation were regularly mentioned during the consultation—notably, risks of peripheral facial paralysis, poor functional outcomes, additional surgeries, post-operative dizziness, and risks of recurrence and/or residual issues (between 40% and 78% of “systematically” responses).

However, five risks were mentioned less frequently: cerebral abscess and meningitis, aesthetic sequelae, taste disturbances, delayed healing and tinnitus (only between 1% and 22% of “systematically” responses).

**Table 4 jcm-13-05651-t004:** Complications Risk Information Following Surgery. For each question, multiple choices were possible. Percentage calculated based on the number of participants. Percentages > 50% are in bold. Situation: only if the situation is appropriate.

Questions	Never N (%)	Rarely N (%)	Situation N (%)	Often N (%)	Systematically N (%)
Facial paralysis	2 (2%)	2 (2%)	16 (20%)	14 (17%)	**51 (62%)**
Poor functional outcome	1 (1%)	1 (1%)	8 (10%)	12 (15%)	**63 (77%)**
New intervention	1 (1%)	3 (4%)	8 (10%)	13 (16%)	**60 (73%)**
Vertigo	8 (10%)	18 (22%)	13 (16%)	18 (22%)	33 (40%)
Tinnitus	8 (10%)	30 (37%)	14 (17%)	15 (18%)	16 (19%)
Aesthetic sequelae	28 (34%)	35 (43%)	14 (17%)	6 (7%)	1 (1%)
Recurrence, residual	1 (1%)	0 (0%)	7 (9%)	10 (12%)	**64 (78%)**
Taste disturbances	16 (20%)	32 (39%)	7 (9%)	12 (15%)	18 (22%)
Meningitis, deep abscess	30 (37%)	21 (26%)	18 (22%)	8 (9%)	8 (9%)
Delayed healing, dehiscence	28 (34%)	25 (30%)	14 (17%)	9 (11%)	6 (7%)

### 3.6. Associations between Population Characteristics and Survey Results

Participants working in a university hospital were more likely to inform patients about the risks of dizziness (*p* = 0.04, chi-squared test), aesthetic sequelae (*p* = 0.04, chi-squared test), poor functional outcomes (*p* = 0.04, chi-squared test), the need for a new intervention (*p* = 0.05, chi-squared test) and peripheral facial paralysis (*p* = 0.05, Fisher’s exact test) but not the risks of delayed healing or dehiscence (*p* = 0.3, chi-squared test), tinnitus (*p* = 0.7, chi-squared test), recurrence or residual (*p* = 0.2, chi-squared test), taste disturbances (*p* = 0.3, chi-squared test) and meningitis and deep abscess (*p* = 0.08, chi-squared test).

Surgeons mainly practicing otology were more likely to inform patients about the risks of cholesteatoma recurrence and/or residual (*p* = 0.02, Fisher’s exact test) and taste disturbances (*p* = 0.02, Fisher’s exact test) but not the risks of facial paralysis (*p* = 0.6, chi-squared test), a poor functional outcome (*p* = 0.6, chi-squared test), a new intervention (*p* = 0.3, chi-squared test), vertigo (*p* = 0.7, chi-squared test), tinnitus (*p* = 0.8, chi-squared test), aesthetic sequelae (*p* = 0.9, chi-squared test), meningitis and deep abscess (*p* = 0.2, chi-squared test) and delayed healing or dehiscence (*p* = 0.5, chi-squared test).

No association was found between the provision of the information form to patients and the geographical area of practice (*p* = 0.5, chi-squared test), age (*p* = 0.07, chi-squared test), gender (*p* = 0.7, chi-squared test), professional structure (*p* = 0.3, chi-squared test), type of professional activity (*p* = 0.9, chi-squared test) and duration of professional practice (*p* = 0.3, chi-squared test). 

No association was found between age and information about risks of facial paralysis (*p* = 0.3, chi-squared test), a poor functional outcome (*p* = 0.2, chi-squared test), a new intervention (*p* = 0.7, chi-squared test), vertigo (*p* = 0.5, chi-squared test), tinnitus (*p* = 0.9, chi-squared test), aesthetic sequelae (*p* = 0.4, chi-squared test), recurrence or residual (*p* = 0.9, chi-squared test), taste disturbances (*p* = 0.8, chi-squared test), meningitis and deep abscess (*p* = 0.8, chi-squared test) and delayed healing or dehiscence (*p* = 0.9, chi-squared test).

No association was found between gender and information about the risks of facial paralysis (*p* = 0.4, chi-squared test), a poor functional outcome (*p* = 0.8, chi-squared test), a new intervention (*p* = 0.6, chi-squared test), vertigo (*p* = 0.4, chi-squared test), tinnitus (*p* = 0.8, chi-squared test), aesthetic sequelae (*p* = 0.9, chi-squared test), recurrence or residual (*p* = 0.9, chi-squared test), taste disturbances (*p* = 0.9, chi-squared test), meningitis and deep abscess (*p* = 0.7, chi-squared test) and delayed healing or dehiscence (*p* = 0.9, chi-squared test).

No association was found between the geographic area and information about the risks of facial paralysis (*p* = 0.9, chi-squared test), a poor functional outcome (*p* = 0.8, chi-squared test), a new intervention (*p* = 0.4, chi-squared test), vertigo (*p* = 0.8, chi-squared test), tinnitus (*p* = 0.9, chi-squared test), aesthetic sequelae (*p* = 0.2, chi-squared test), recurrence or residual (*p* = 0.9, chi-squared test), taste disturbances (*p* = 0.4, chi-squared test), meningitis and deep abscess (*p* = 0.2, chi-squared test) and delayed healing or dehiscence (*p* = 0.6, chi-squared test).

No association was found between the participants’ years of practice and information about the risks of facial paralysis (*p* = 0.2, chi-squared test), a poor functional outcome (*p* = 0.7, chi-squared test), a new intervention (*p* = 0.9, chi-squared test), vertigo (*p* = 0.4, chi-squared test), tinnitus (*p* = 0.9, chi-squared test), aesthetic sequelae (*p* = 0.5, chi-squared test), recurrence or residual (*p* = 0.8, chi-squared test), taste disturbances (*p* = 0.9, chi-squared test), meningitis and deep abscess (*p* = 0.9, chi-squared test) and delayed healing or dehiscence (*p* = 0.9, chi-squared test).

Multivariable analyses could not be conducted due to the associations between patient characteristics and the risk of an adverse outcome having a *p*-value greater than 0.2, except for the previously mentioned results.

## 4. Discussion

This study aimed to describe the French professional clinical practices during preoperative consultations for cholesteatoma removal surgery in 2024. The various questions were designed to assess as broadly as possible the oral and written information given to patients, ensuring fully informed consent before the surgery. This study emphasizes the need for better communication strategies in preoperative settings to enhance patient understanding and facilitate informed consent.

The information regarding cholesteatoma pathology, the surgical procedure and its modalities was largely well explained based on the survey results. The patients received good information, with a written information form provided in 83% of cases, and in 75% of cases, this form was validated by a professional society.

However, the information provided about surgical risks varied significantly, with limited details on risks such as cerebral involvement, tinnitus, aesthetic risks, taste disturbances and delayed healing. The risk of intracranial complications was systematically discussed in only 9% of cases, despite its severity [21,22,23]. Some of these less-discussed risks, such as tinnitus, can contribute to postoperative patient discomfort [24]. In view of these results, and to improve comprehensive information during preoperative consultations for cholesteatoma removal surgery in France, an information form for this surgery validated by the French Society of Oto-Rhino-Laryngology and Head and Neck Surgery could be developed, describing all potential complications and the surgical process. This could provide standardized information in addition to oral information to improve patient comprehension and surgical outcomes as well as improve communication with patients and the confidence relationship with the surgeon. Educational video support could also help patients better understand surgery (e.g., via a QR code on the information form).

Providing comprehensive information about surgical risks, along with written support, is crucial to ensuring a trustworthy physician–patient relationship and reducing postoperative legal actions. These principles align with the concept of providing “loyal, clear, and appropriate information on the state of health, investigations, and proposed treatments”, as outlined in the French Public Health Code (Article R4127-35). Informed consent is a legal obligation and a potential source of litigation. For example, in Japan, 3% of litigations were related to a lack of informed consent [16]. In the United States, this figure could rise to 22% based on a study of 30 years of lawsuits following otologic surgery [17]. Among these cases, 48% were post-mastoidectomy, 21% were post-ossiculoplasty and 16% were post-myringoplasty [17]. The most common postoperative complications leading to litigation were hearing loss (in 45% of cases) and peripheral facial paralysis (in 38% of cases). In 31% of cases, the verdict favored the patient, with an average damage award of USD 1,131,189 per patient [17]. 

Patient satisfaction holds significant importance to medical practices and remains challenging to address in the management of cholesteatoma. Sixty-three percent of participants do not inquire about the patient’s expectations regarding surgery. While the patient may be cured of their condition, the lack of postoperative benefits can be perceived as a surgical failure. Further investigation into postoperative patient satisfaction and the factors influencing it would be valuable. In the United States, Americans regularly assess their practitioners based on the Patient Satisfaction score (PS score). This method of evaluating physicians by patients has also been perceived as harmful in the field of otology [25]. External factors influencing the PS score include wait times, geographic localization, patient ethnicity or age [26]. A study carried out in India, involving over 4000 patients with ENT conditions, showed patients’ satisfaction levels in university hospital settings compared to those in general hospitals. This could be attributed to access difficulties and specialized consultations leading to communication challenges between physicians and patients [27].

To improve patient information and satisfaction, a strategy could be introducing preoperative paramedical support. Utilizing an advanced practice nurse to address supplementary questions alongside medical consultations would be beneficial. This approach has undergone testing in the ENT and anesthesia departments and demonstrated improvements [28,29,30].

A number of projects could follow up on this first survey to explore several lines of research in greater depth. It would indeed have been interesting to know the healthcare practices or resources of the different French otologists, even if no association was found between the different results of the survey and the geographical area. While not included in the current survey, this aspect could be explored in future research to provide a more comprehensive analysis. The influence of cholesteatoma size and location on the information given to patients, particularly the risks of complications, warrants further study.

One of the limitations of this work is recruitment bias, despite our efforts to minimize it by directly distributing the surveys to the professional email addresses of university otology professors. Additionally, the survey was published in an SFORL newsletter and on an internet platform for ENT (Ear, Nose and Throat) specialists. Indeed, 62% of the participants worked in a university hospital, while only 25% of the 2996 French ENT specialists work in public hospitals (both university hospitals and non-university hospitals). We may wonder if ENT specialists working in university hospitals could be more interested in responding to surveys related to clinical research.

Another limitation of our work is the low number of respondents among French ENT specialists. Unfortunately, data on the number of French otologists performing cholesteatoma surgery are unavailable, which must be smaller and more in accordance with our target population. We cannot establish a real and precise response rate to the survey because we do not know the number of surgeons who had access to it via the various distribution channels. Another French study published in 2023 concerning voice rehabilitation post-total laryngectomy had included 75 respondents in their survey, which is slightly lower than our number of respondents, even if they are not the same pathologies [31].

We aimed to compare French preoperative clinical practices for cholesteatoma removal surgery with those of other countries to increase the study’s relevance and impact. However, to our knowledge, no prior research has explored this subject in the scientific literature.

## 5. Conclusions

In this study, we showed that cholesteatoma removal surgery was well explained to the patient during the preoperative consultation, mostly with written support. Information appears to be more precise when surgeons perform otologic surgeries more regularly or work in a university hospital. Our results highlight the potential benefit of developing an information form dedicated to this surgery, complemented by computed tools to show educational videos.

## Figures and Tables

**Figure 1 jcm-13-05651-f001:**
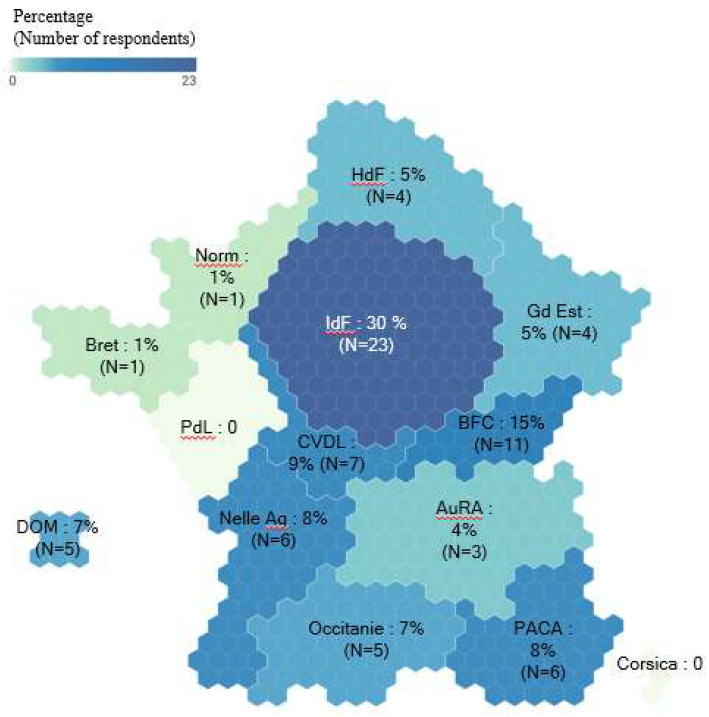
Geographical distribution of participants (n = 82). HdF: Hauts-de-France Region; Norm: Normandy Region; IdF: Île-de-France Region; Gd Est: Grand Est Region; Bret: Brittany Region; PdL: Pays de la Loire Region; CVDL: Centre-Val-de-Loire Region; BFC: Burgundy-Franche-Comté Region; AuRa: Auvergne-Rhône-Alpes Region; Nelle Aq: Nouvelle-Aquitaine Region; Occ: Occitanie Region; PACA: Provence-Alpes-Côte d’Azur Region; Corse: Corsica Region; DOM: Overseas Region.

**Table 1 jcm-13-05651-t001:** Population Characteristics.

Description	N (%)
Gender: - Male - Female	47 (57%) 35 (44%)
Age: - Under 40 years - 40–49 years - 50–59 years - Over 60 years	37 (45%) 16 (19%) 14 (17%) 15 (18%)
Structure: (multiple choices possible) - University hospital - General hospital - Semi-public - Private - Salaried employee	51 (62%) 15 (18%) 7 (8%) 15 (18%) 0 (0%)
Activity: - Mainly otological - Equally distributed among subspecialties - Not focused on otology - No surgical otology, but sent to a corresponding specialist - No otological activity	42 (52%) 26 (32%) 12 (15%) 1 (1%) 0 (0%)
Number of surgeries performed per year: - Over 40 (>1×/week) - Between 20 and 40 (Approx. 1×/week) - Between 10 and 20 (1 to 2×/month) - Less than 10 (<1/month)	20 (24%) 11 (13%) 21 (26%) 30 (37%)
Years of practice: - Less than 5 years - Between 5 and 10 years - Between 10 and 20 years - Between 20 and 30 years - Over 30 years	18 (21%) 23 (28%) 12 (14%) 15 (18%) 15 (18%)

## Data Availability

The dataset is available on request from the authors. The raw data supporting the conclusions of this article will be made available by the authors on request. Please send your requests to the address of the corresponding author: caroline.guigou@chu-dijon.fr.

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
