# Peer review of "A French Preoperative Cholesteatoma Management: Current Preoperative Consultation and Tendencies"

_jcm, 2024, doi:10.3390/jcm13185651_

Round 1
Reviewer 1 Report (New Reviewer)
Comments and Suggestions for Authors
Summary of the Paper
The manuscript titled "A French preoperative cholesteatoma management: current preoperative consultation and tendencies" investigates the practices surrounding preoperative consultations for cholesteatoma surgery in France. The study involved a survey of 82 otolaryngology surgeons, focusing on their approaches to informing patients about the surgery, associated risks, and postoperative expectations. Key findings indicate variability in how surgeons communicate surgical information, with some not providing formal information forms or adequately discussing potential risks. The authors suggest that developing standardized information tools and educational resources could enhance patient understanding and improve surgical outcomes. The study emphasizes the need for better communication strategies in preoperative settings to ensure patients are well-informed about their procedures.
Reviewer's Comment
The manuscript presents a timely and relevant investigation into the preoperative management of cholesteatoma, addressing an important aspect of patient care in otolaryngology. The methodology is sound, and the results provide valuable insights into current practices among surgeons. The authors effectively highlight the variability in communication and the potential benefits of standardized information tools.
Minor points:
1. The authors should have provided the information and the results of the open-ended question.
2. I suggest the authors clarify the rationale behind the survey questions and provide more detailed recommendations for implementing the proposed information tools.
3. A more thorough discussion of the implications of their findings on clinical practice would strengthen the manuscript.
This study contributes significantly to the field and can improve patient outcomes in cholesteatoma management.
Comments on the Quality of English LanguageMinor editing of English language required.
Author Response
Dear reviewer,
All the co-authors would like to thank you for the quality of your questions and your kind comments.
We hope we have answered your questions correctly.
We remain at your entire disposal if necessary.
Best regards.
Minor points:
Q.1. The authors should have provided the information and the results of the open-ended question.
A.1 Thank you for your remark. The “open-ended question” is stated line 106 in the paragraph “Materials and Methods”.
“only” has been added in this sentence: “The open-ended question: "Why do you not provide an information form about the surgery?" which was asked only if participants answered negatively to the question about giving an information form to the patient during the preoperative consultation”.
The answers of the open-ended question are already detailed in results line 188: “When no information form was provided (13% of cases, 11 participants) (answers of the open-ended question), the reasons were: the information form was provided by non-medical staff in 1 case, the feeling that the form was unnecessary or useless in 7 cases, inadequacy of the form with the practice's specificity in 1 case, and 2 participants did not specify their reason”.
The information that these are the answers to the open-ended question has been added between brackets in this sentence.
Q.2. I suggest the authors clarify the rationale behind the survey questions and provide more detailed recommendations for implementing the proposed information tools.
A.2 To clarify the rationale behind the survey questions, this sentence has been added line 284: “The various questions were designed to assess as broadly as possible the oral and written information given to patients, ensuring fully informed consent before the surgery. This study emphasizes the need for better communication strategies in preoperative settings to enhance patient understanding and facilitate informed consent.”
In view of the results of our work, a dedicated information form and video material could help standardize the information given to patients. Now theses sentences have been added (or modified) in the discussion line 297: “In view of these results and to improve comprehensive information during preoperative consultations for cholesteatoma removal surgery in France, an information form for this surgery validated by the French Society of Oto-Rhino-Laryngology and Head and Neck Surgery could be developed, describing all potential complications and the surgical process. This could provide standardized information in addition to oral information to improve patient comprehension and surgical outcomes, as well as improving communication with patients and the confidence relationship with the surgeon. Educational video support could also help patients to better understand surgery (e.g. via a QR code on the information form).”
Q.3. A more thorough discussion of the implications of their findings on clinical practice would strengthen the manuscript.
A.3 Thank you very much for you comment. Now theses sentences have been added (or modified) in the discussion line 297: “In view of these results and to improve comprehensive information during preoperative consultations for cholesteatoma removal surgery in France, an information form for this surgery validated by the French Society of Oto-Rhino-Laryngology and Head and Neck Surgery could be developed, describing all potential complications and the surgical process. This could provide standardized information in addition to oral information to improve patient comprehension and surgical outcomes, as well as improving communication with patients and the confidence relationship with the surgeon. Educational video support could also help patients to better understand surgery (e.g. via a QR code on the information form).”
Q.4 This study contributes significantly to the field and can improve patient outcomes in cholesteatoma management.
A.4 Thank you very much.
Reviewer 2 Report (New Reviewer)
Comments and Suggestions for Authors
Abstract:
The abstract gives a broad overview of the study but doesn't go into enough detail about its main components. Here are some ideas for improvement:
- Clearly state the study's purpose or objective. The current statement is extremely general, attempting "to describe the French professional clinical practices..."
- Give more information about the sample population, methodology, and study design. Mention that the study was a survey, together with the number of participants, the administration method, and any important inclusion or exclusion criteria.
- Provide a summary of the main findings from the quantitative survey and any related statistical analysis. Don't forget to emphasize one or two of the most significant or noteworthy discoveries.
- Include a summary of the study's relevance and the implications of the results in the conclusion.
Introduction:
Although it gives a solid overview of cholesteatoma and stresses the value of preoperative knowledge, the introduction may be improved:
- Describe the specific knowledge gaps this study attempts to address and its justification. Why was this survey research necessary right now? Which major clinical question or questions does it aim to address?
- Add extra references to help put the issue in context and emphasize the goals.
Give a clear explanation of the study objectives and questions before delving into the methodology section.
Methods:
The methods section clearly describes the survey development and administration but lacks some key details:
- Describe the timeframe in detail: when and for how long was the survey conducted?
- Explain the procedure used to create the survey. Was it in any manner piloted or put through validity/reliability tests?
- Give more information about the sample population, including the number of people invited, the number of people who responded, and the response rate. To what extent was the sample representative?
- Give an explanation for the selection of specific statistical tests and support the conducted analysis.
Results:
The results give a comprehensive overview of the survey replies; nevertheless, they might be strengthened by adding: - More quantitative information;
- Giving specific figures and percentages for the most important outcomes.
- Rather of merely summarizing important findings in text, use more figures and tables.
- Don't merely state the results verbatim; instead, highlight the most noteworthy and pertinent discoveries in the text.
- Report findings from statistical analysis as outlined in the methodology.
Discussion:
The discussion is underdeveloped and lacking interpretation of the results:
- Consider the major findings in light of previous research's knowledge. Contrast and compare with other pertinent studies.
- Talk about the study's sample, analytic, and methodological constraints to support the findings.
- Go into further detail about the findings' importance. What are the main conclusions and how do they relate to clinical practice?
any
Author Response
Dear reviewer,
All the co-authors would like to thank you for the quality of your questions.
We hope we have answered your questions correctly.
We remain at your entire disposal if necessary.
Best regards.
Abstract:
Q.1 The abstract gives a broad overview of the study but doesn't go into enough detail about its main components.
A.1 Now, the abstract has been reworked following your advice. We've tried to include as much detail as possible, bearing in mind that the maximum word count is set by the editor at 250 words.
Q.2 Here are some ideas for improvement:
Clearly state the study's purpose or objective. The current statement is extremely general, attempting "to describe the French professional clinical practices..."
A.2 Now, the aims were defined as follows in the abstract:
“This study aimed to characterize the information delivery during preoperative consultations for cholesteatoma removal surgery in 2024. The secondary objective was to identify any factors influencing the information delivered.” (line 10)
Q.3 Give more information about the sample population, methodology, and study design. Mention that the study was a survey, together with the number of participants, the administration method, and any important inclusion or exclusion criteria.
A.3 Now, this sentence has been added line 12: “this study was a practice survey”.
In the results, it is noted that 82 surgeons responded to the survey (line 17).
For inclusion or exclusion criteria, theses sentences have been added in the “material and methods” section: “All respondents to the survey were included. There were no exclusion criteria.” (line 86)
Q.4 Provide a summary of the main findings from the quantitative survey and any related statistical analysis. Don't forget to emphasize one or two of the most significant or noteworthy discoveries.
A.4 Thank you for your remark.
We have now added a sentence to show the disparity of information given depending on the type of complication: “The risk of recurrence or residual post-operative cholesteatoma was systematically stated in 78% of cases (n=64), while the risk of aesthetic sequelae was only stated in 1% (n=1).” (line 19)
In addition, statistically significant results are already included in the summary. We remain at your disposal if you think that other relevant results should be added to the abstract.
Q.5 Include a summary of the study's relevance and the implications of the results in the conclusion.
A.5 Now, the conclusion had been modified as follow: “Cholesteatoma surgery was well explained to patients during the preoperative consultation, mostly with written support, even if the information given was not the same for all complication risks. It could be useful to create an information form dedicated to cholesteatoma surgery, to improve comprehensive information and maintain a trustworthy relationship with patients.” (line 24)
Introduction:
Although it gives a solid overview of cholesteatoma and stresses the value of preoperative knowledge, the introduction may be improved:
Q.6 Describe the specific knowledge gaps this study attempts to address and its justification. Why was this survey research necessary right now? Which major clinical question or questions does it aim to address?
A.6 In the introduction, we described the importance of providing information prior to surgery, so that patients can give informed consent. We explained the importance of the information form, which could be given in consultation (without obligation), and the risk of litigation if there are post-operative complications without clear information given beforehand. That's where our study comes in. When we carried out a bibliography on this subject, very few studies were interested in how information was provided during pre-operative consultations for cholesteatoma removal surgery. In France, we have no information form dedicated to this type of surgery. That's why we decided to carry out this study.
Now, these sentences have been added into the introduction: “We aimed to investigate the information provided during preoperative consultations for cholesteatoma removal surgery in France, as there was no standardized information form. To address this, we developed a survey to evaluate the information provided to patient in the preoperative consultation ensuring they can give informed consent.” (line 71)
Q.7 Add extra references to help put the issue in context and emphasize the goals.
Give a clear explanation of the study objectives and questions before delving into the methodology section.
A.7 We'd like to thank you for your helpful comment. It would indeed be interesting to introduce our study by citing other studies that have carried out surveys on patients' pre-operative information. To the best of our knowledge, we have found no other studies on this subject. We've pointed this out in the introduction and also in the discussion, because it's a limitation of our work: we have no means of external comparison (lines 70 and 364).
Now, the objectives have been redefined as follows: “The main objective of this study aimed to describe the French professional clinical practices regarding the information delivery during preoperative consultations for cholesteatoma removal surgery in 2024 through data collected by a nationally distributed survey. The secondary objective was to identify any factors that could influence the information given to patients.” (line 76)
Methods:
The methods section clearly describes the survey development and administration but lacks some key details:
Q.8 Describe the timeframe in detail: when and for how long was the survey conducted?
A.8 As indicated in line 91, the survey was created and distributed October 2023 and March 2024.
Q.9 Explain the procedure used to create the survey. Was it in any manner piloted or put through validity/reliability tests?
A.9 Thank you for your comment. The study was created using Cherries recommendations. No validity/reliability tests have been performed.
Q.10 Give more information about the sample population, including the number of people invited, the number of people who responded, and the response rate. To what extent was the sample representative?
A.10 Thank you very much for this comment, which makes sense. As indicated on line 88, we used 3 distribution ways to get as many respondents as possible. We asked otology professors to distribute the survey in their respective regions. We don't have access to the number of subscribers to the SFORL newsletter, nor to the number of subscribers to the internet platform for ENT. We therefore have no way of knowing how many surgeons had access to the survey, or of establishing a real and precise response rate.
In the discussion (line 356), we criticize the low number of respondents to our work in relation to the total number of French ENT specialists. We have added this sentence to the discussion: “We cannot establish a real and precise response rate to the survey because we do not know the number of surgeons who had access to it via the various distribution channels.” (line 359)
Q.11 Give an explanation for the selection of specific statistical tests and support the conducted analysis.
A.11 The statistical tests were selected and performed by a statistician, Dr Serge Aho-Ludwig, who is co-author for this work. We feel that we have correctly detailed the statistics in the materials and methods section. We remain at your disposal if you have any specific questions about the statistics.
Results:
The results give a comprehensive overview of the survey replies; nevertheless, they might be strengthened by adding: -
Q.12 More quantitative information;
A.12 Thank you for your comment. However, we can't add more quantitative information as all the data we have is already in the results.
Q.13 Giving specific figures and percentages for the most important outcomes.
A.13 Thank you for your comment. However, we can't add more percentages as all the data we have is already in the results. We've chosen to present our results in table form rather than in figures (except for figure 1) for greater readability.
Q.14 Rather of merely summarizing important findings in text, use more figures and tables.
A.14 We're really sorry, but we're having difficulty answering your question. We have used 1 figure and 4 tables to present our results.
We remain at your entire disposal should you wish to specify more precisely which results you would like to see in additional tables or figures.
Q.15 Don't merely state the results verbatim; instead, highlight the most noteworthy and pertinent discoveries in the text.
A.15 Thank you for your comment. The most relevant information is written before the tables in sub-paragraphs 3.3, 3.4 and 3.5. The order of the text has been changed in paragraph 3.6 so that the significant associations are visible. We hope we have answered your question correctly.
Q.16 Report findings from statistical analysis as outlined in the methodology.
A.16 Thank you for your comment, which gives the article a good methodology. As we mentioned earlier, all the statistics were done by a statistician. All our results are in line with the statistics set out in the materials and methods section.
Discussion:
The discussion is underdeveloped and lacking interpretation of the results:
Q.17 Consider the major findings in light of previous research's knowledge. Contrast and compare with other pertinent studies.
A.17 Thank you very much for your comment. As indicated in a previous comment, we have not found any work that has studied the information given to patients during pre-operative consultations. We have therefore insisted during the discussion on the importance of information given to limit litigations, increase the confidence relationship with the surgeon. This lack of bibliographic references is part of the limitations of our work and is discussed at the end of the discussion.
We have also reworked the discussion to better highlight the clinical applications of this work
Q.18 Talk about the study's sample, analytic, and methodological constraints to support the findings.
A.18 The limits of our work are already described in the discussion between lines 350 and 365. This is mainly the distribution channels of the survey and the number of respondents. We remain available if you find that there are other limitations or biases in our work.
Q.19 Go into further detail about the findings' importance. What are the main conclusions and how do they relate to clinical practice?
A.19 Thank you very much for you comment.
Now theses sentences have been added (or modified) in the discussion line 297 for the clinical implications: In view of these results and to improve comprehensive information during preoperative consultations for cholesteatoma removal surgery in France, an information form for this surgery validated by the French Society of Oto-Rhino-Laryngology and Head and Neck Surgery could be developed, describing all potential complications and the surgical process. This could provide standardized information in addition to oral information to improve patient comprehension and surgical outcomes, as well as improving communication with patients and the confidence relationship with the surgeon. Educational video support could also help patients to better understand surgery (e.g. via a QR code on the information form).
Reviewer 3 Report (New Reviewer)
Comments and Suggestions for Authors
The authors provided an interesting summary of the otological practice of patient preparation that takes place in France. Noteworthy is the small number of respondents, which the authors noted as limitations of the work. The introduction, description of methods and results as well as the discussion are conducted efficiently and described in an understandable way. The table entry needs clarification:
- What is the difference in structer (table 1): Private and Employed private - should be explaind.
Only 6 of the 18 references items are from the last 5 years.
Author Response
Dear reviewer,
All the co-authors would like to thank you for the quality of your questions and your kind comments.
We hope we have answered your questions correctly.
We remain at your entire disposal if necessary.
Best regards.
Q.1 What is the difference in structer (table 1): Private and Employed private - should be explaind.
A.1 Thank you for your comment and sorry for the confusion. “Employed private” has been replaced by “Salaried employee”.
Q.2 Only 6 of the 18 references items are from the last 5 years.
A.2 Thank you very much for your comment. You are right. As far as we know, we haven't found any more recent references or other publications dealing with this subject. In fact, we discussed it at the end of line 364.
This manuscript is a resubmission of an earlier submission. The following is a list of the peer review reports and author responses from that submission.
Round 1
Reviewer 1 Report
Comments and Suggestions for Authors
Dear authors, dear Editor;
The present study "A French preoperative cholesteatoma management: current clinical practices and tendencies" aims to describe the professional clinical practices in France during preoperative consultations for cholesteatoma surgery in the year 2024. The study used a survey distributed by the French Society of Oto-Rhino-Laryngology and Head and Neck Surgery to collect data from 82 surgeons. The main contributions of the paper are its comprehensive overview of current preoperative practices, highlighting the use of written information forms and the detailing of surgical risks and procedures.
Major comments:
- Ethic committee statement is missing. If there is no ethics approval available, a proof of waiver by the committee (+translation in englisch) is obligated.
- Data availability statement is missing
- How many ENT surgeons work in france? it would be interesting to know if the aquired sample size of 89 patients is a representative sample size. Involvement of a professional statistician would be valuable.
- the article mentions the intent to compare French practices with those of other countries but does not provide much comparisons. Including international benchmarks or discussing differences with practices in other countries could enhance the study's relevance and impact.
- please provide always the exact p-value, not just p>0.05.
- The geographical distribution of participated surgeons is informative. However, providing additional context about regional differences in healthcare practices or resources would add depth to the analysis.
Author Response
Reviewer 1:
Dear reviewer,
Thank you very much for the time you spent evaluating our work and the quality of your comments.
We hope you will find our answers satisfactory.
Should you have any further comments or questions, please do not hesitate to contact us.
Best regards.
Major comments:
Q.1: Ethic committee statement is missing. If there is no ethics approval available, a proof of waiver by the committee (+translation in english) is obligated.
A.1: Thank you for your remark. This study is a practice survey. It was performed in accordance with the principles of good clinical practice and the need for Ethics Committee approval was waived. ENT were informed and consented to their participation in this study. After examination of this study, this trial is outside Jardé’s law field.
Now, theses sentences have been added line 77.
A proof of waiver by the committee is attached in pdf format.
Q.2: Data availability statement is missing.
A.2: We apologize for this omission. Now, a paragraph concerning the Data availability statement has been added line 362: “The data set is available on request from the authors. The raw data supporting the conclusions of this article will be made available by the authors on request. Please send your requests to the address of the corresponding author: caroline.guigou@chu-dijon.fr”.
Q.3: How many ENT surgeons work in France? it would be interesting to know if the aquired sample size of 89 patients is a representative sample size. Involvement of a professional statistician would be valuable.
A.3: Thank you very much for your comment. This is indeed one of the limitations of our work that we haven't developed enough in the discussion.
As indicated line 334 in the discussion, 2,996 ENT specialists are currently working in France. 89 respondents to our questionnaire is indeed small in relation to the total number of ENT specialists. It is important to note that not all ENT specialists perform cholesteatoma removal surgery. We do not have the number of French otologists at our disposal.
On the other hand, another French study published in 2023 concerning voice rehabilitation post total laryngectomy had included 75 respondents to their survey, which is slightly lower than our number of respondents even if they are not the same pathologies (Al Burshaid et al., Acta Otolaryngol, 2023).
As the number of respondents is one of the limitations of our work, a dedicated paragraph has been added to the discussion line 338: “Another limitation of our work is the low response rate among French ENT specialists. Unfortunately, data on the number of French otologists performing cholesteatoma surgery is unavailable, which must be smaller and more in accordance with our target population. Another French study published in 2023 concerning voice rehabilitation post total laryngectomy had included 75 respondents to their survey, which is slightly lower than our number of respondents even if they are not the same pathologies (Al Burshaid et al., Acta Otolaryngol, 2023)”.
All the statistics in this article were provided by a professional statistician, Dr Serge Aho-Ludwig, who co-authored our work.
Q.4: The article mentions the intent to compare French practices with those of other countries but does not provide much comparisons. Including international benchmarks or discussing differences with practices in other countries could enhance the study's relevance and impact.
A.4: We agree with you. We would have compared our results with those of other countries to get a more international perspective.
As indicated in line 344 in the discussion, we have not found any other article in the scientific literature that has evaluated the good clinical practices of ENT specialists during the pre-operative consultation for cholesteatoma removal.
The paragraph in the discussion concerning this subject has been modified as follows to support this point: “We aimed to compare French preoperative clinical practices for cholesteatoma removal surgery with those of other countries to increase the study's relevance and impact. However, to our knowledge, no prior research has explored this subject in the scientific literature.”
Q.5: please provide always the exact p-value, not just p>0.05.
A.5: Your attention is appreciated. All p values are now integrated into the text.
Several paragraphs have been added to the “Associations between population characteristics and survey results” section (line 219).
Q.6: The geographical distribution of participated surgeons is informative. However, providing additional context about regional differences in healthcare practices or resources would add depth to the analysis.
A.6: Thank you for your comment, which highlights another limitation of our work. For your information, all French regions have a university hospital center. It would indeed have been interesting to know the healthcare practices or resources of the different French otologists, even if no association was found between the different results of the survey and the geographical area. This was not asked for in the survey, and could be the subject of a future study.
Now, this new paragraph has been added in the discussion line 322: “A number of projects could follow up on this first survey to explore several lines of research in greater depth. It would indeed have been interesting to know the healthcare practices or resources of the different French otologists, even if no association was found between the different results of the survey and the geographical area. While not included in the current survey, this aspect could be explored in future research to provide a more comprehensive analysis.”

Reviewer 2 Report
Comments and Suggestions for Authors
In cases where cholesteatoma is in its early stage and hearing is still very good, regular observation may be chosen over surgery. However, it is clear that the standard treatment for COM with cholesteatoma is surgery. Especially when the involvement of cholesteatoma is extensive, the risks of hearing loss, facial paralysis, and taste disturbances naturally increase. These complications can be present before surgery or may arise as a result of the surgery. Such complications significantly impact the patient's quality of life, so there is no doubt about the importance of preoperative counseling regarding these issues.
However, providing appropriate counseling and explanation is not as easy as it seems. There is significant variation depending on the physician giving the explanation, and the emphasis on certain aspects can vary greatly depending on the extent of the patient’s cholesteatoma involvement. Therefore, the efforts of the French Society of Oto-Rhino-Laryngology and Head and Neck Surgery (SFORL) to standardize explanations and guidance are highly commendable.
Title
The content I expected from reading the title differs quite a bit from the actual content of the article. From the title, I anticipated an overall view of the current state of cholesteatoma management in French medical institutions. For example, I expected to see statistics on the proportion of cases where surgical treatment is recommended for early-stage cholesteatoma. However, the article mainly focuses on the explanations given before surgery. In reality, ‘clinical practice’ encompasses a broader scope, including decision-making processes about surgery, not just preoperative consultations. Although the authors specified that the scope is limited to ‘preoperative cholesteatoma management,’ I believe the term ‘clinical practice’ is not appropriate. It might be better to simply specify it as ‘preoperative consultation.’
Introduction
“To improve patient information, the French Society of Oto-Rhino-Laryngology and Head and Neck Surgery (SFORL) has published over thirty informational forms for patients that surgeons can provide alongside consultations. These forms offer a written, consensual, and formalized support detailing the procedure, postoperative modalities, and specific risks of each surgery.” Have these guidelines already been published? If so, is the ultimate purpose of this study to evaluate whether preoperative consultations are being conducted well according to these guidelines? Or is the purpose to further improve these guidelines? At the very least, it would be helpful for the readers of this study to have access to the informational form.
Materials and Methods
The scope of preoperative counseling inevitably varies significantly depending on the extent of cholesteatoma involvement. There will naturally be differences in the level of explanation between congenital cholesteatoma and a huge cholesteatoma involving damage to the lateral semicircular canal and facial nerve canal. How were these differences reflected in the survey?
Results & Discussion
It appears that physicians working in university hospitals provide more detailed explanations about the complications. This is likely because these physicians are the ones who actually perform the surgeries. Instead of categorizing by the type of institution where they work, I believe it would be more appropriate to distinguish between physicians who perform the surgeries and those who do not. In my opinion, the study should be conducted exclusively with physicians who actually perform the surgeries. It might be better to exclude responses from other physicians, as including them could skew the results. The authors have also mentioned this sampling bias. Excluding physicians who do not perform the surgeries would likely address this issue of sampling bias more effectively.
Comments on the Quality of English LanguageSome of the sentences are a bit awkward. Most of them are fine.
Author Response
Dear reviewer,
Thank you very much for the time you spent evaluating our work and the quality of your comments.
We hope you will find our answers satisfactory.
Should you have any further comments or questions, please do not hesitate to contact us.
Best regards.
Title
Q.1: The content I expected from reading the title differs quite a bit from the actual content of the article. From the title, I anticipated an overall view of the current state of cholesteatoma management in French medical institutions. For example, I expected to see statistics on the proportion of cases where surgical treatment is recommended for early-stage cholesteatoma. However, the article mainly focuses on the explanations given before surgery. In reality, ‘clinical practice’ encompasses a broader scope, including decision-making processes about surgery, not just preoperative consultations. Although the authors specified that the scope is limited to ‘preoperative cholesteatoma management,’ I believe the term ‘clinical practice’ is not appropriate. It might be better to simply specify it as ‘preoperative consultation.’
A.1: Thank you for your comment, which makes the title more accurate. Now 'clinical practice' has been replaced by ‘preoperative consultation’.
Introduction
Q.2: “To improve patient information, the French Society of Oto-Rhino-Laryngology and Head and Neck Surgery (SFORL) has published over thirty informational forms for patients that surgeons can provide alongside consultations. These forms offer a written, consensual, and formalized support detailing the procedure, postoperative modalities, and specific risks of each surgery.” Have these guidelines already been published? If so, is the ultimate purpose of this study to evaluate whether preoperative consultations are being conducted well according to these guidelines? Or is the purpose to further improve these guidelines? At the very least, it would be helpful for the readers of this study to have access to the informational form.
A.2: Thank you for your comment. These are not guidelines but informational forms that the ENT can provide in consultations. ENT are not obliged to issue this form to patients. This form is not in itself compulsory. There is no legal provision for it, nor any obligation to provide it. This information form is part of a “good practice” recommended to satisfy the duty to inform set out in the French Public Health Code (Art. L1111-2 and R4127-35) and Code 35 of medical ethics. This notion of the duty to inform differs from one specialty to another, depending on the current state of the art, and simply enables judges to verify, in the event of a liability claim, whether the patient has given informed consent, or whether the doctor has committed a fault (in the civil sense) by failing to inform the patient sufficiently. This information form does not replace the information given orally during the consultation, but provides the patient with a written record, enabling him or her to reflect after the consultation.
These forms have never been the subject of scientific publications. These documents are freely available at: https://campusorl.fr/public/interventions-chirurgicales-en-orl/.
This information is added line 53 in the introduction.
Our study was a descriptive work. We wanted to have a vision of the professional practices of the ENT during consultations before a cholesteatoma removal surgery in 2024, especially if the SFORL information forms were given to patients and not to create an information form dedicated to cholesteatoma, even though we think it would be interesting to do so to improve the information given to patients (line 299 in the discussion).
Materials and Methods
Q.3 The scope of preoperative counseling inevitably varies significantly depending on the extent of cholesteatoma involvement. There will naturally be differences in the level of explanation between congenital cholesteatoma and a huge cholesteatoma involving damage to the lateral semicircular canal and facial nerve canal. How were these differences reflected in the survey?
A.3: Thank you for your comment. Obviously, the surgical risk is not the same depending on the size and location of the cholesteatoma. Nevertheless, all surgical risks must be explained before any surgery, in order to provide patients with the clearest and most complete information.
We have not evaluated the impact of cholesteatoma size and location on the information given about the risks of complications in our work. This could be the subject of a future questionnaire. In the discussion, a paragraph was added as follows line 322: “A number of projects could follow up on this first survey to explore several lines of research in greater depth. It would indeed have been interesting to know the healthcare practices or resources of the different French otologists, even if no association was found between the different results of the survey and the geographical area. While not included in the current survey, this aspect could be explored in future research to provide a more comprehensive analysis. Investigating the influence of cholesteatoma size and location on the information given to patients, particularly the risks of complications warrant further study.”
Results & Discussion
Q.4: It appears that physicians working in university hospitals provide more detailed explanations about the complications. This is likely because these physicians are the ones who actually perform the surgeries. Instead of categorizing by the type of institution where they work, I believe it would be more appropriate to distinguish between physicians who perform the surgeries and those who do not. In my opinion, the study should be conducted exclusively with physicians who actually perform the surgeries. It might be better to exclude responses from other physicians, as including them could skew the results. The authors have also mentioned this sampling bias. Excluding physicians who do not perform the surgeries would likely address this issue of sampling bias more effectively.
A.4: Thank you for your comment. Indeed, in the study of the association between population characteristics and questionnaire results, it was shown that:
“Participants working in a university hospital were more likely to inform patients about the risks of dizziness (p=0.04, chi-squared test), aesthetic risks (p=0.04, chi-squared test), poor functional outcomes (p=0.04, chi-squared test), need for a new intervention (p=0.05, chi-squared test), and risk of peripheral facial paralysis (p=0.05, Fisher's exact test). Surgeons mainly practicing otology were more likely to inform patients about the risks of cholesteatoma recurrence and/or residual issues (p=0.02, Fisher's exact test) and taste disturbances (p=0.02, Fisher's exact test).” (line 226).
ENT specialists working in university hospitals are not necessarily those who perform the most cholesteatoma removal surgery (p=0.1, chi-squared test). This information is now added to the results (line 156).
Also, in Table 1, which groups together the characteristics of respondents, only 1 ENT (1%) does not perform surgery. With the exception of the latter, all respondents perform cholesteatoma removal surgery. This shows that, even with a population of surgeons only, those working in a university hospital are still very present among our respondents. This is why we have mentioned it in the discussion as a potential bias.
Round 2
Reviewer 1 Report
Comments and Suggestions for Authors
the manuscript has been sufficiently improved
Author Response
Dear reviewer,
Thank you very much for all your work and your answer.
Best regards
Reviewer 2 Report
Comments and Suggestions for Authors
The manuscript was much improved and I understand the authors' intentions better.
Comments on the Quality of English LanguageThere don't seem to be any quality issues of English.